# Extreme sensitivity to ultraviolet light in the fungal pathogen causing white-nose syndrome of bats

Jonathan M. Palmer [1], Kevin P. Drees[2], Jeffrey T. Foster [2,3] & Daniel L. Lindner [1]

Bat white-nose syndrome (WNS), caused by the fungal pathogen *Pseudogymnoascus destructans*, has decimated North American hibernating bats since its emergence in 2006. Here, we utilize comparative genomics to examine the evolutionary history of this pathogen in comparison to six closely related nonpathogenic species. *P. destructans* displays a large reduction in carbohydrate-utilizing enzymes (CAZymes) and in the predicted secretome (~50%), and an increase in lineage-specific genes. The pathogen has lost a key enzyme, UVE1, in the alternate excision repair (AER) pathway, which is known to contribute to repair of DNA lesions induced by ultraviolet (UV) light. Consistent with a nonfunctional AER pathway, *P. destructans* is extremely sensitive to UV light, as well as the DNA alkylating agent methyl methanesulfonate (MMS). The differential susceptibility of *P. destructans* to UV light in comparison to other hibernacula-inhabiting fungi represents a potential "Achilles' heel" of *P. destructans* that might be exploited for treatment of bats with WNS.

[1] Center for Forest Mycology Research, Northern Research Station, US Forest Service, Madison, WI 53726, USA. [2] Department of Molecular, Cellular, and Biomedical Sciences, University of New Hampshire, Durham, NH 03824, USA. [3]Present address: Pathogen and Microbiome Institute, Northern Arizona University, Flagstaff, AZ, USA. Correspondence and requests for materials should be addressed to D.L.L. (email: dlindner@fs.fed.us)

White-nose syndrome of bats (WNS) continues to decimate hibernating bat populations in North America. The fungal disease was first documented in 2006 in eastern North America (New York) and the fungus has advanced west across the continent, most recently being detected on the West coast of North America[1]. WNS can result in >90% mortality in local hibernating bat populations[2] and is caused by the psychrophilic fungus *Pseudogymnoascus destructans*[3,4]. The fungus has a strict temperature growth range of ~4–20 °C and therefore can only infect bats during hibernation[5]. WNS is not a systemic infection, but rather is characterized by *P. destructans* colonization of the skin of hibernating bats, which manifests as cupping skin erosions based on histopathology[6]. Frequent arousal from hibernation, depletion of fat reserves, and dehydration appear to contribute to mortality in infected individuals[7–10]. *P. destructans* has been found throughout Eurasia and occasionally causes mild WNS symptoms; however, no mass mortality events have been observed in Eurasia[11]. *P. destructans* has spread in a "bulls eye" pattern in North America and has only been found in environments where WNS-infected bats are found, strongly suggesting that the fungus is not native to North America and represents a classic example of an introduced pathogen decimating a naïve population[12,13]. *P. destructans* is a member of the under-studied Pseudeurotiaceae family; while recent studies have resulted in several draft genome assemblies within this group, much of the biology of the fungal pathogen and its relatives remains unknown.

Here, we present functionally annotated genomes for *P. destructans*, as well as six closely related nonpathogenic *Pseudogymnoascus* species. Comparison of the pathogen with the nonpathogenic species provides an opportunity to obtain insight into the origins and adaptations of the fungal pathogen of WNS.

## Results

**Sequencing, assembly, and annotation**. Recent phylogenetic work that aimed at identification and resolution of closely related species to *P. destructans* resulted in moving this species from genus *Geomyces* to *Pseudogymnoascus*[4]. Moreover, sampling for fungal isolates from hibernacular soil resulted in identification of other closely related *Pseudogymnoascus* species that are not known to be pathogenic[12]. We previously described the hybrid genome assembly of *P. destructans*[14]; in addition, we chose six closely related species and sequenced these isolates using Illumina chemistry, resulting in ~×250 coverage for each genome (strains listed in Supplementary Table 1). All seven *Pseudogymnoascus* genomes were within the expected genome size for haploid ascomycetes (~30–36 MB) with ~50% GC content (Table 1). Genome annotation was completed using funannotate v0.1.8 (https://github.com/nextgenusfs/funannotate), resulting in 9335 protein-coding genome models for *P. destructans* and a range of 10,252–11,033 protein gene models for the nonpathogenic *Pseudogymnoascus* species (Table 1). Genome functional annotation

(see "Methods" section) was added to each gene model; complete functional annotation is available in Supplementary Data 1–7.

**Orthology and evolutionary phylogeny**. Due to the importance of WNS, as well as interest in cold-tolerant fungi, there has recently been several draft genomes sequenced in the genus *Pseudogymnoascus*. To delineate relationships between these genomes, we acquired all *Pseudogymnoascus* genomes deposited in NCBI, as well as several outgroup species: *Aspergillus nidulans*, *Botrytis cinerea*, *Fusarium fujikuroi*, *Penicillium chrysogenum*, and *Neurospora crassa* (Supplementary Table 2). Single-copy BUSCO orthologs[15] were extracted using Phyloma (https://github.com/nextgenusfs/phyloma) and were used to generate a maximum likelihood phylogeny in RAxML v8.29 (PROTGAMMALG; 1000 bootstrap replicates) from 822 concatenated gene models. To estimate time of evolutionary divergence, node calibrations from Beimforde et al.[16] were used in r8s v1.80[17]; Leotiomycetes–Sordariomycetes 267–430 MYA, Eurotiomycetes 273–537 MYA, Sordariomycetes 207–339 MYA, and Pezizomycotina 400–583 MYA (Supplementary Data 8). This analysis illustrates that the nonpathogenic *Pseudogymnoascus* species sequenced here are among the closest known relatives of *P. destructans* (Fig. 1). Additionally, these data suggested that the last known common ancestor of *P. destructans* diverged approximately 23.5 MYA. The oldest known chiropteran species in the fossil record, *Palaeochiropteryx*, was estimated to have lived 50–40 MYA[18,19], while the most recent adaptive speciation of Eurasian *Myotis* bat species occurred 9–6 MYA and the North American *Myotis* species emerged more recently at 6–3.2 MYA[20]. Thus, Eurasian *Myotis* species were present at the time when *P. destructans* diverged from its relatives, suggesting that it could have coevolved alongside modern-day Eurasian chiropteran species.

ProteinOrtho5[21] was used to identify 3949 single-copy orthologous groups between seven *Pseudogymnoascus* species used in this study, representing only 42.3% of the protein-coding genes in *P. destructans*. Relationships among these species were inferred from a maximum likelihood RAxML phylogeny based on the concatenated alignment of 500 orthologous proteins using *Botrytis cinerea* as an outgroup (Fig. 2). The orthologous proteome analysis identified 1934 unique proteins in *P. destructans* that were not found in any of the nonpathogenic *Pseudogymnoascus* species (Fig. 2 and Table 1), making *P. destructans* the species with the most unique proteins in our study. To identify orthologous proteins under positive selection, dN/dS ratios for all orthologous groups were calculated using the codeml M0 model from PAML[22]. Likelihood ratio tests comparing the M1/M2 models, as well as M7/M8 models with significance at <0.05 were calculated to validate the estimated dN/dS ratios greater than 1, ratios that suggest positive selection is occurring[22]. Forty-six orthologous groups displayed evidence of positive selection. Of those orthologous groups under positive

**Table 1 Genome summary statistics of *P. destructans* and nonpathogenic *Pseudogymnoascus* species**

| Species | Isolate | Assembly size | Largest scaffold | Average scaffold | Num scaffolds | Scaffold N50 | Percent GC | Repetitive DNA | Num genes | Num proteins | Num tRNA | Unique proteins |
|---|---|---|---|---|---|---|---|---|---|---|---|---|
| *P. destructans* | 20631-21 | 35,818,201 | 2,552,699 | 431,545 | 83 | 1,168,637 | 49.24% | 38.17% | 9575 | 9335 | 240 | 1934 |
| *P.* sp. 23342-1-I1 | 23342-1-I1 | 32,900,320 | 990,412 | 33,640 | 978 | 289,626 | 49.94% | 7.35% | 10,914 | 10,762 | 152 | 1275 |
| *P.* sp. 24MN13 | 24MN13 | 30,179,533 | 129,019 | 10,675 | 2827 | 24,078 | 50.17% | 3.45% | 10,368 | 10,269 | 99 | 1090 |
| *P.* sp. WSF3629 | WSF3629 | 35,517,105 | 741,106 | 86,839 | 409 | 189,864 | 48.85% | 10.78% | 11,193 | 11,033 | 160 | 1053 |
| *P.* sp. 3VT5 | 03VT05 | 33,292,640 | 361,480 | 38,893 | 856 | 115,508 | 49.09% | 15.94% | 10,393 | 10,252 | 141 | 795 |
| *P.* sp. 5NY8 | 05NY08 | 32,206,663 | 867,888 | 56,403 | 571 | 205,172 | 49.74% | 10.22% | 10,669 | 10,514 | 155 | 480 |
| *P. verrucosus* | UAMH10579 | 30,174,856 | 1,768,408 | 199,833 | 151 | 446,342 | 50.36% | 4.60% | 10,715 | 10,573 | 142 | 515 |

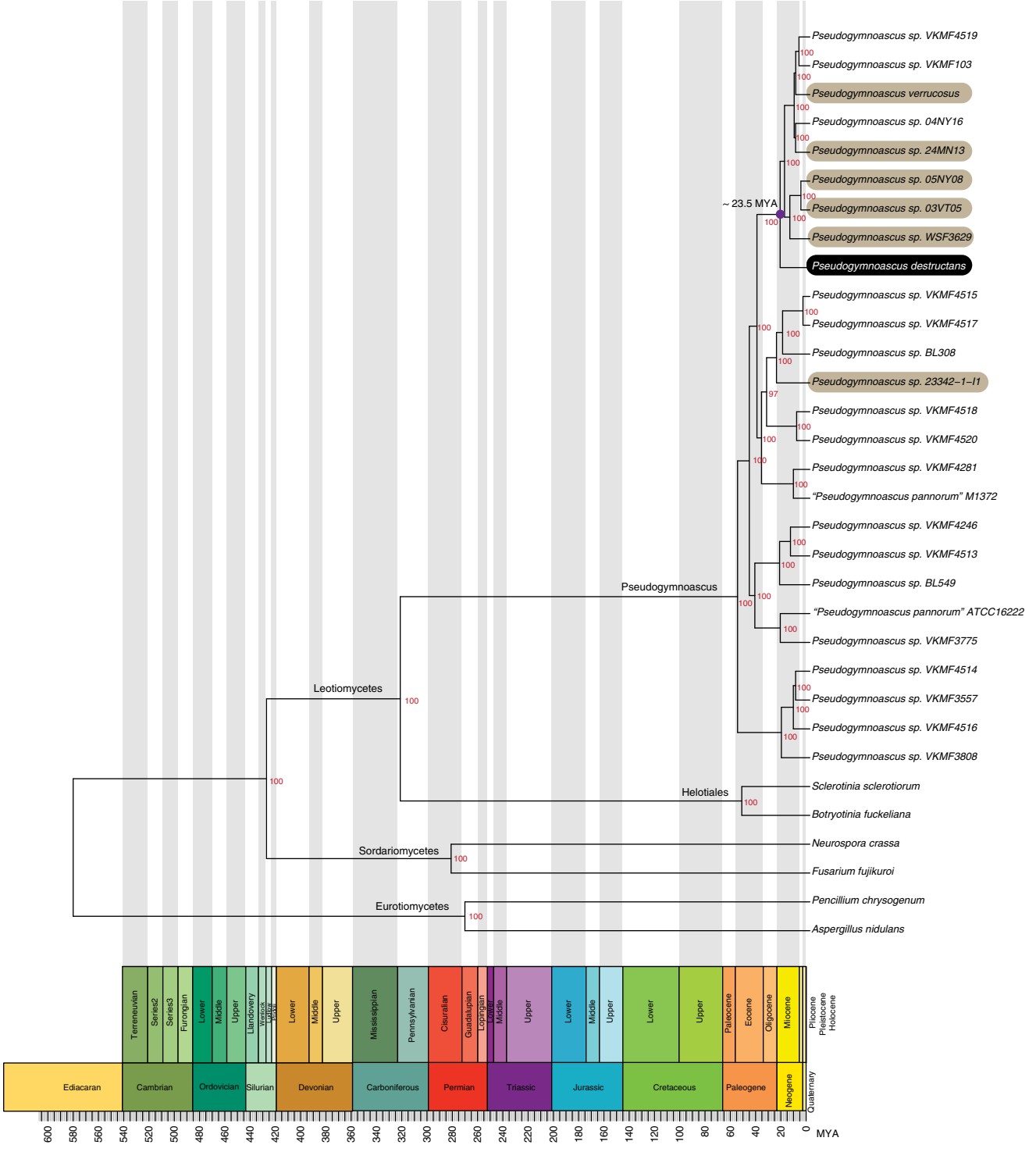

**Fig. 1** Maximum likelihood phylogeny of *Pseudogymnoascus* draft genomes and several outgroup Ascomycetes calibrated using fossil evidence. The six nonpathogenic *Pseudogymnoascus* species sequenced here are among the closest known relatives of *P. destructans*. *P. destructans* was estimated to have diverged from its last common ancestor around 23.5 MYA. Node support values are derived from 1000 bootstrap replicates

selection, only 14 contained a protein from *P. destructans* (Supplementary Data 9). However, functional annotation for these orthologous groups did not provide sufficient information to predict a function for any of these orthologs.

**Genome-level comparisons.** Carbohydrate-activating enzymes (CAZymes) are a group of proteins involved in the breakdown and/or utilization of carbon. Genomes of fungi associated with plants, either plant pathogens or decomposers, generally harbor a greater number of CAZymes than those causing disease in animals[23]. The six nonpathogenic *Pseudogymnoascus* species on average harbored 493 CAZymes, ranging from 463 to 544; however, the genome of *P. destructans* contains only ~36% of the average number of CAZymes (179 in total) for the group (Fig. 2; Supplementary Data 10). The CAZymes can be further broken

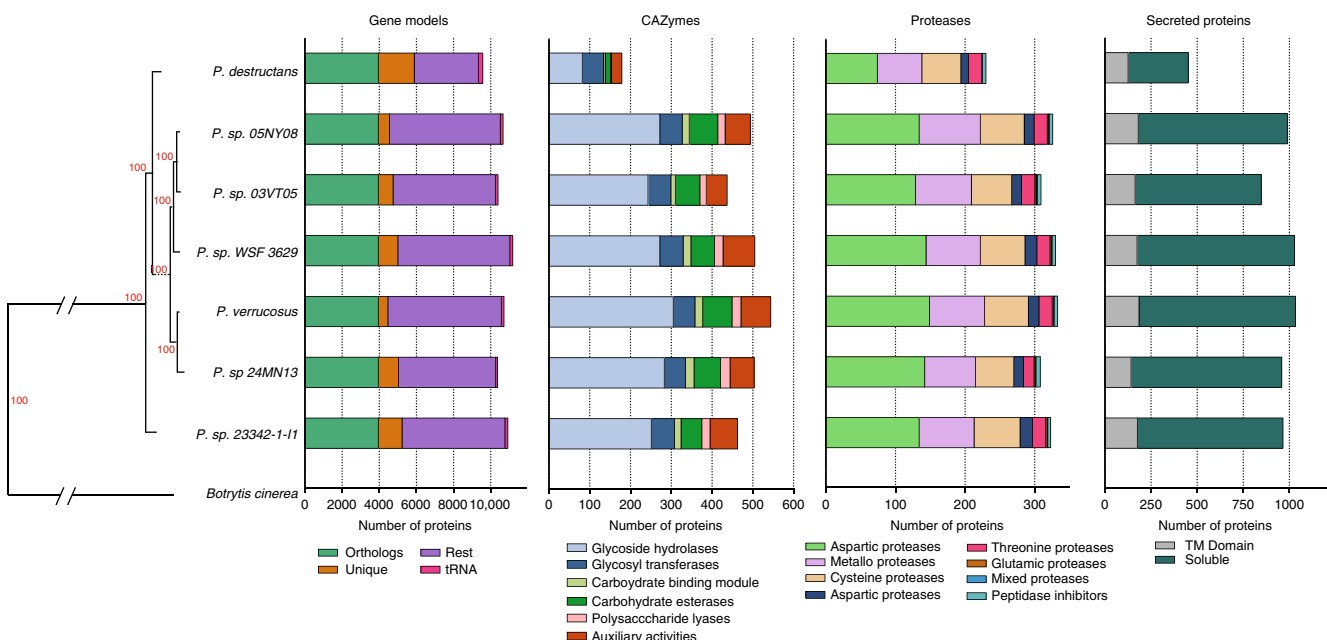

**Fig. 2** Comparative genomic analyses of orthologous proteins of *Pseudogymnoascus destructans* and six close relatives. Maximum likelihood phylogeny illustrates relationships between fungi used in this study. Genome-level comparisons made based on number of gene models, number of CAZymes, number of proteases, and the number of secreted proteins as described in Methods

down into enzyme classes/modules. In comparison to the average of the nonpathogenic *Pseudogymnoascus* species, the genome of *P. destructans* harbors ~69% (22) of auxiliary activity (AA) enzymes, ~17% (4) of proteins containing carbohydrate-binding modules (CBM), ~20% (13) of the carbohydrate esterases (CE), ~30% (82) of the glycoside hydrolases (GH), ~10% (2) of the pectin lyases (PL), and ~95% (52) of the glycosyltransferases (GT). Moreover, there were no specific CAZy families that were expanded in *P. destructans* that would indicate enhanced ability to utilize certain carbohydrate sources. Due to the large decrease in CAZymes, we hypothesized that *P. destructans* would have limited growth on complex and/or different carbon sources in comparison to the nonpathogenic *Pseudogymnoascus* species. To test this hypothesis, we compared growth on 190 carbon sources using a modified Biolog Phenotype Array Platform. Consistent with the large reduction in CAZymes in its genome, *P. destructans* was unable to utilize many carbon sources that the nonpathogenic *Pseudogymnoascus* species could readily utilize (Fig. 3, Supplementary Data 11). These data are consistent with a recent report comparing growth characteristics of *P. destructans* to closely related soil fungi[24]. The reduced CAZyme repertoire is a shared characteristic of other fungal pathogens of animals, including the dermatophytes, the true fungal dimorphic pathogens, as well as skin-inhabiting yeasts[23]. Presumably, the reduction in CAZymes reflects a change in carbon availability/acquisition during the evolution from an ancestral saprobe to a bat pathogen, i.e., *P. destructans* may have shed CAZyme pathways necessary for living in soil/sediments but are not required for growth on hibernating bats.

A group of subtilisin serine endopeptidases (MEROPS family S08A) of *P. destructans* have previously been described as PdSP1, PdSP2,[25] and one of them (PdSP2; destructin-1) was subsequently shown to degrade collagen, suggesting a potential role in colonization of bat skin[26]. Using the MEROPS protease database, 230 predicted proteases were identified from the *P. destructans* genome, which represents a 28% reduction compared to the average of the nonpathogenic *Pseudogymnoascus* species (average: 321; range: 308–330) (Fig. 2; Supplementary Data 12). The

subtilisin proteases appear to be conserved in the *Pseudogymnoascus* species analyzed here and therefore is consistent with the observation from Vraný et al.[27] that collagen-degrading enzymes are present in several nonpathogenic soil microorganisms. While proteases are likely to be involved in colonization of bat skin tissue by *P. destructans*, the previously identified subtilisin proteases (e.g., PdSP2; destructin-1) do not appear to have evolved specifically for this purpose. Moreover, PdSP2 has recently been shown to be more highly expressed in laboratory culture medium than during WNS[28,29], indicating that it may not play a large role during pathogenesis.

Secreted proteins are generally important for fungi as digestion of nutrients using secreted enzymes occurs outside the fungal cell. Recently, fungal effectors have been described in plant pathogens (some of which can enter animal cells), an example of the intricate competitive arms race between pathogen and host (reviewed in refs. [30,31]). The predicted secretome of *P. destructans* is also reduced by more than 50% in comparison to the nonpathogenic *Pseudogymnoascus* species (452 vs. average of 970 secreted proteins; range: 848–1033) (Fig. 2). The trend of fewer secreted proteins in fungal pathogens of animals has been previously described[32,33], with one hypothesis being that losing unnecessary secreted proteins is an evolutionary strategy to evade vertebrate immune systems[32]. Previous reports indicate that fungal secretomes can be lineage specific, consistent with the observation that 56 of the 362 predicted soluble secreted proteins in *P. destructans* (~15%) do not have orthologs in the nonpathogenic *Pseudogymnoascus* species.

To visually depict the functional variation between species, we generated count data of InterProScan Domain and Pfam domains for each genome in our study (Supplementary Data 13–14). The resulting matrices were visualized using a nonmetric multidimensional scaling (NMDS) ordination, which consistently identified *P. destructans* as distinct from the nonpathogenic *Pseudogymnoascus* species (Supplementary Fig. 2). To identify functional groups and processes that were enriched (positively or negatively), which could help explain the shift in functional domains between *P. destructans* and the nonpathogenic

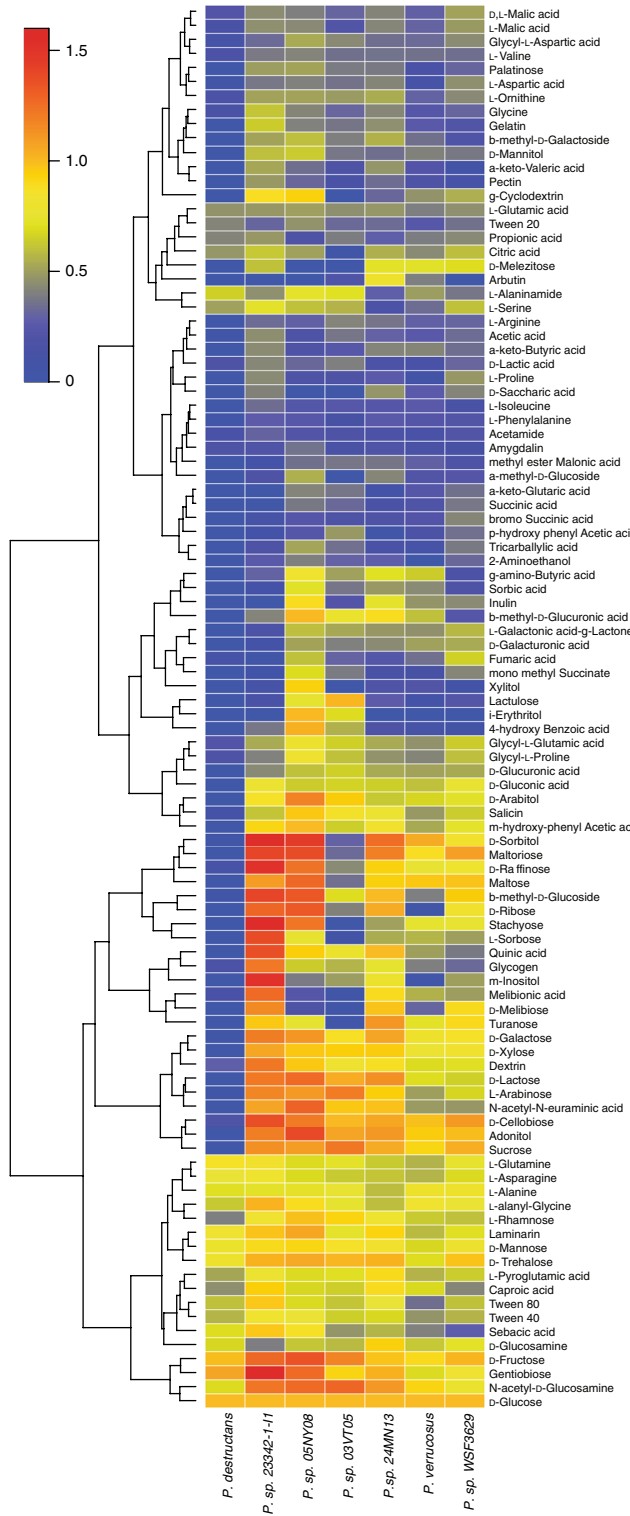

**Fig. 3** Heatmap of Biolog phenotypic microarrays testing fungal growth. Carbon utilization from 190 sources was tested to assess the growth of each of seven *Pseudogymnoascus* species. Growth was quantified using ImageJ ColonyArea plug-in after 15 °C incubation for 7 days for the nonpathogenic *Pseudogymnoascus* species and after 14 days for *P. destructans*. Growth for each carbon source is presented as normalized to growth on glucose. These data are derived from a single biological replicate (*n* = 1)

*Pseudogymnoascus* species, we used a Gene Ontology (GO) term enrichment method. As expected and consistent with the ordinations, no enrichment of GO terms was found for any of the six nonpathogenic *Pseudogymnoascus* species. Two very broad biological processes (BP) were enriched in *P. destructans*: cellular processes (GO:0009987) and cellular metabolic processes (GO:00044327) (Supplementary Table 3). On the other hand, some more specific BP terms were underrepresented in *P. destructans*: transmembrane transport (GO:0055085), carbohydrate metabolic process (GO:005975), and oxidation–reduction process (GO:0055114). This is consistent with reductions in the secretome, as well as CAZymes using alternate methods as previously discussed. Additionally, the GO enrichment identified a trend in underrepresented BP terms related to transcription (GO:0006351, GO:0097659, and GO:0032774) (Supplementary Table 3). Generally, these data suggest that major genome-level differences between *P. destructans* and the nonpathogenic *Pseudogymnoascus* species are driven by the large differences in carbohydrate utilization enzymes, as well as a reduction in the putative secretome.

A hallmark of many ascomycete fungi is their ability to produce bioactive small molecules, such as the pharmaceuticals lovastatin and penicillin[34]. These small molecules are also known as secondary metabolites and many are produced by the coordinated effort of gene clusters (genes physically located in proximity to each other on a chromosome). Secondary metabolites have been hypothesized to be involved in niche exploitation by fungi, including evasion of vertebrate immune systems[35]. AntiSMASH v3.0[36] was used to predict secondary metabolite gene clusters from each genome. Based on antiSMASH prediction, *P. destructans* harbors 14 predicted secondary metabolite gene clusters, while the average for the nonpathogenic *Pseudogymnoascus* species was 27 gene clusters, with a range of 14–36 (Supplementary Table 4, Supplementary Data 15–21). Despite having fewer putative secondary metabolism clusters, the *P. destructans* genome encodes for the two non-ribosomal-polyketide synthetase enzymes required to produce the iron-scavenging siderophores that have previously been characterized chemically: ferrichrome and triacetylfusarinine C[37] (Supplementary Data 15). Additionally, *P. destructans* harbors a putative melanin cluster that has been shown to be involved in pathogenicity in other fungi such as the opportunistic human pathogen *Aspergillus fumigatus*; four of the six nonpathogenic *Pseudogymnoascus* species also harbor a similar melanin-like cluster, which in *A. fumigatus* is physically located in the cell wall of spores of the fungus and has been hypothesized to function as protection from ultraviolet (UV) light and reactive oxygen species (ROS) damage[38,39]. The secondary metabolite arsenal of the Pseudeurotiacae fungi in this study is less than some other groups such as the Eurotiomycetes, which can have 70 or more secondary metabolism clusters. However, several of the secondary metabolites produced by *P. destructans* warrant future study as they could contribute to the development and persistence of WNS in bats.

An interesting characteristic of the *P. destructans* genome is that it contains a large expansion of repetitive DNA sequences, accounting for 38.17% of the genome, a significant expansion in comparison to the nonpathogenic *Pseudogymnoascus* species (Table 1). Coinciding with this repeat expansion is a reduction in total gene models, as well as a general trend toward a reduction in several gene families, driven largely by a massive reduction in CAZymes, secreted proteins, and proteases. At the same time, *P. destructans* contains more lineage-specific genes than the nonpathogenic *Pseudogymnoascus* species (Fig. 2). The observation of expanded repetitive elements and increasing numbers of lineage-specific genes is reminiscent of the lineage-

specific regions/chromosomes in *Fusarium oxysporum*[40]. However, in contrast to *F. oxysporum*, analysis of the repetitive sequences across the *P. destructans* genome suggests a uniform distribution (Supplementary Fig. 2A) and lineage-specific genes were not colocalized with repetitive regions (Supplementary Fig. 2B). Similarly to fungal pathogens in the Onygenales, *P. destructans* lacks proteins containing the fungal cellulose-binding domain (CBM18)[41]; however, this does not seem to be tied to pathogenicity as 5 of the 6 nonpathogenic *Pseudogymnoascus* species also lack this functional domain. We were unable to identify protein families that were expanded in *P. destructans*, somewhat reminiscent of *Coccidioides*, which have very few expanded protein families[41]. This is in stark contrast to the amphibian chytrid pathogens (*Batrachochytrium dendrobatidis* and *Batrachochytrium salamandrivorans*), which have undergone an expansion of protease gene families, as well as their genomes en route to pathogenicity[42]. Given the evolutionary distance between the chytrids and ascomycete fungi, it is perhaps not surprising that these devastating fungal pathogens have different evolutionary trajectories.

**Light and DNA repair pathways.** Taken together, our data suggest that *P. destructans* has been a pathogen of bats for millions of years and thus has likely coevolved in the absence of light. Most organisms that have been found in the absence of light maintain the ability to repair DNA caused by UV light radiation, including most microbes that have been isolated from hibernacula[43,44]. Moreover, a metagenomics analysis of a cave ecosystem identified an overrepresentation of DNA repair enzymes, despite the absence of UV light[45]. We challenged the fungal species studied here with four different DNA-damaging agents that included UV, methyl methanesulfonate (MMS), 4-nitroquinoline (4-NQO), and camptothecin (CPT). While we observed slight differences in sensitivity among species (and even between isolates of *P. destructans*) with 4-NQO and CPT, the most dramatic differential sensitivity was seen with UV light and MMS (Fig. 4a). To further characterize the sensitivity to UV light, we employed a quantitative conidial survival assay and exposed the fungi to three different wavelengths of UV light and three different exposure levels (Fig. 4b). These data show that none of the fungi tested were sensitive to UV-A (366 nm) light at the dosage tested, while *P. destructans* is differentially sensitive to the higher-energy UV-B (312 nm) and UV-C (254 nm). Remarkably, a low dose of 5 mJ/cm$^2$ exposure of UV-C light resulted in only ~15% survival, while a 10 mJ/cm$^2$ UV-C exposure resulted in <1% survival of *P. destructans* (Fig. 4b).

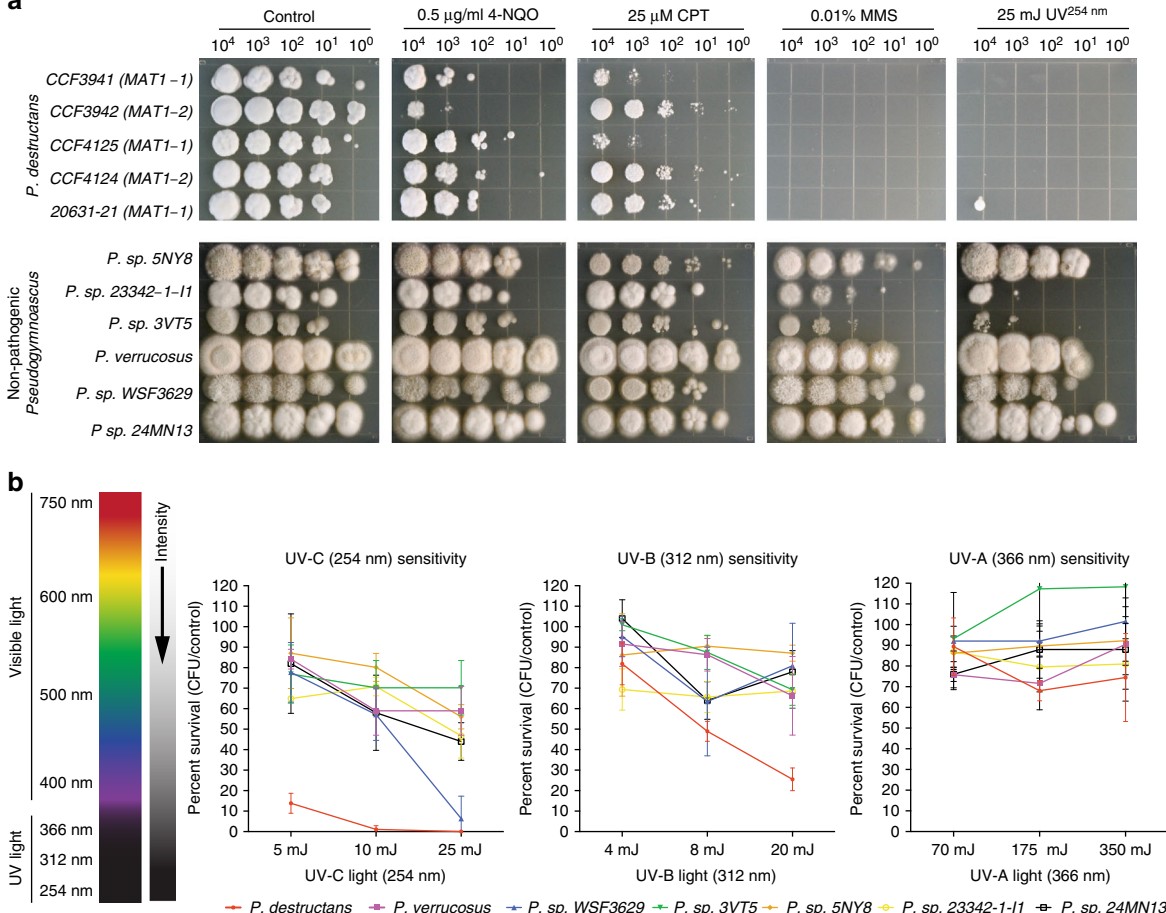

**Fig. 4** Sensitivity of *Pseudogymnoascus* species to DNA-damaging agents. **a** Qualitative plate assay measuring the effects of four DNA-damaging agents on the growth of each fungal species (4-nitroquinoline (4-NQO), camptothecin (CPT), methyl methanesulfonate (MMS), and 254-nm ultraviolet light (UV-C)). Fungal spores were serially diluted, inoculated on appropriate medium, and growth was quantified after 7-day incubation at 15 °C. Multiple isolates of *P. destructans* were tested alongside the nonpathogenic *Pseudogymnoascus* species. **b** A quantitative assay using colony-forming units (CFUs) to measure survival of each fungus under different wavelengths of UV light (254 nm (UV-C), 312 nm (UV-B), and 366 nm (UV-A)). CFU assays were conducted in biological triplicate ($n = 3$) and error bars represent standard deviations

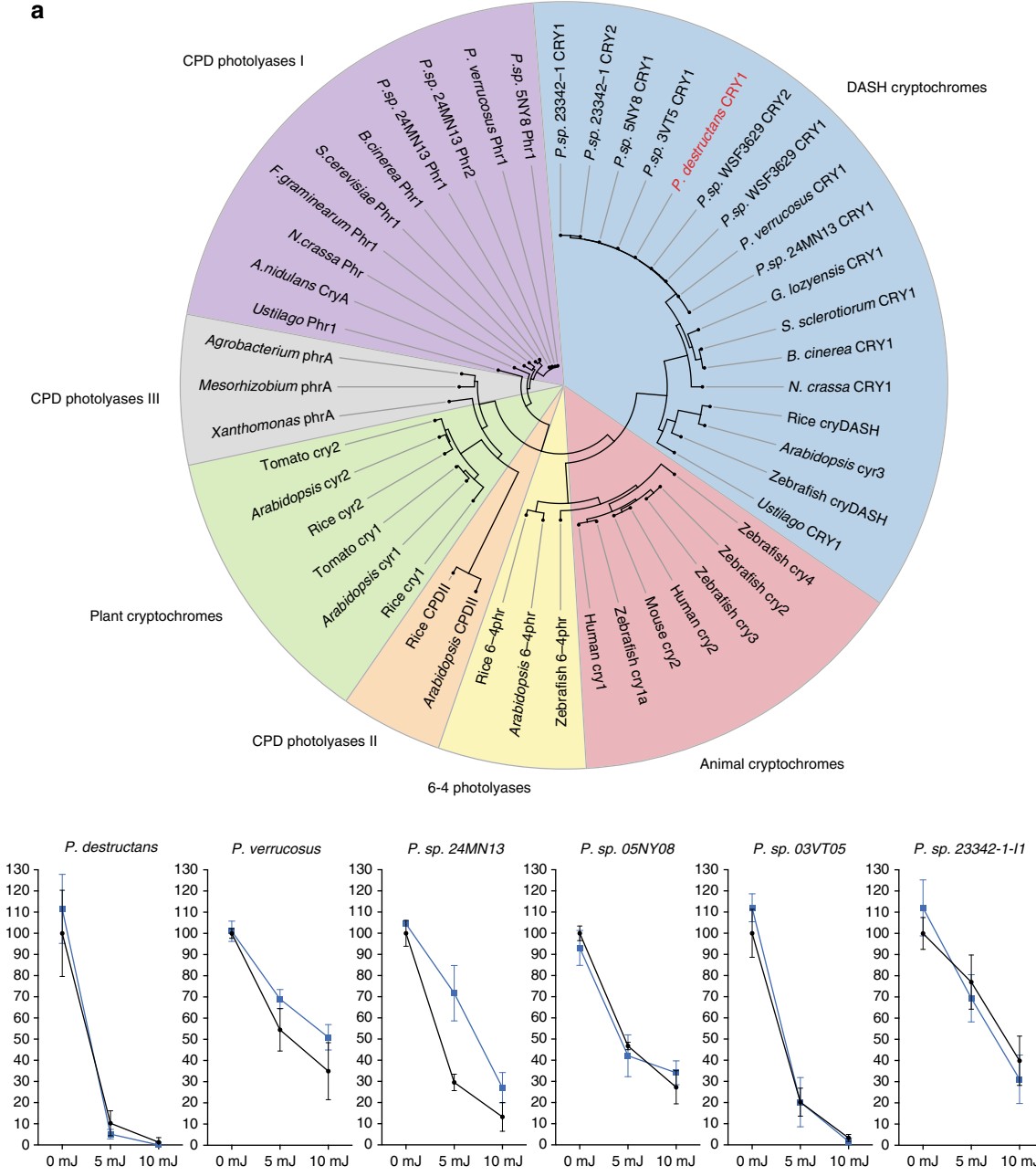

**Fig. 5** Blue light-mediated photoreactivation DNA repair is not functional in *P. destructans*. **a** Characterization of photolyases in *Pseudogymnoascus* species based on protein alignment and maximum likelihood phylogeny of well-characterized enzymes. *Pseudogymnoascus destructans* lacks a canonical CPD photolyase I, although it harbors a cyr-DASH photolyase (VC83_00225). The genomes of *Pseudogymnoascus verrucosus*, *P. sp. 24MN13*, and *P. sp. 05NY08* contain a CPD photolyase I, whereas all species studied contain at least one cry-DASH ortholog. Proteins identified in ref. [66] were used for phylogenetic comparison of DNA photolyases. **b** Photoreactivation experiments comparing survival of germinating conidia exposed to varying doses of UV-C (254 nm) followed by treatment with UV-A (366 nm) for 1 h indicate that only *P. verrucosus* and *P. sp. 24MN13* display increased survival attributed to the activity of CPD photolyase I. No change in survival was detected in *P. destructans* or the other three nonpathogenic *Pseudogymnoascus* species tested. Interestingly, *P. sp. 05NY08* did not show an increase in CFU survival despite the fungus harboring a CPD photolyase I enzyme. A similar phenomenon occurs in *Aspergillus nidulans*, where there was no change in CFU survival in photoreactivation assays in wild-type background, despite the presence of the functional CryA gene (CPD photolyase I)[67]. CFU assays were performed in biological triplicate (*n* = 3) and error bars represent standard deviation

UV light damages DNA by inducing the formation of pyrimidine dimers (cyclobutane dimers and 6-4 photoproducts), while MMS alkylates guanine and adenine nucleotides[46]. Damaged or modified nucleotides result in mispairing and/or replication fork blockage and therefore can result in mutations. Camptothecin's mode of action is similar as it inhibits topoisomerase I function, which results in stalled replication forks; thus, lethal mutations can accumulate during S phase[47]. On the other hand, 4-NQO induces single base pair mutations with a bias toward guanine to thymine transversions[48] and thus can cause direct mutagenesis. To combat DNA damage, organisms have employed several DNA repair pathways, including

photoreactivation, base excision repair (BER), nucleotide excision repair (NER), double-strand break repair, mismatch repair, as well as alternate excision repair (AER) (reviewed in refs. [49–51]). Using fission yeast (*Schizosaccharomyces pombe*) as a model organism, we extracted 169 proteins annotated as involved in DNA repair from www.pombase.org and queried them against the *Pseudogymnoascus* proteomes using BLAST (Supplementary Data 22). Six putative hits, UVE1, MAG1, MAG2, CENP-X, SWI5, and PIF1, were identified on the basis that they were absent in *P. destructans* but present in most of the six nonpathogenic *Pseudogymnoascus*. To validate that orthologous proteins were in fact missing as opposed to being misannotated, we queried the corresponding Pfam hidden Markov model (HMM) profiles from (http://www.pfam.org/) using an exhaustive HMM model search in Phyloma (https://github.com/nextgenusfs/phyloma), which can identify truncated and/or unannotated genes. These results indicated that MAG1, MAG2, and CENP-X orthologs were missing from all *Pseudogymnoascus* genomes (which is consistent with the low scores of the BLAST hits), while partial matches were found for SWI5 and PIF1, indicating either missed gene models during annotation or nonfunctional proteins. Finally, we were unable to find any trace of UVE1 in the *P. destructans* genome, despite finding clear homologs in the six nonpathogenic *Pseudogymnoascus* species, suggesting that UVE1 plays a large role in the repair of UV-damaged DNA in *Pseudogymnoascus*.

*P. destructans* harbors a putative DNA photolyase (VC83_00225); thus, photoreactivation could be a mechanism the fungus uses to repair UV-induced DNA lesions. However, DNA photolyases require light for function and therefore are unlikely to be a major contributor to DNA repair in hibernacula. Preliminary testing of survivability of *P. destructans* incubated in either light or dark indicated that light was insufficient to repair DNA lesions (Supplementary Fig. 3). There are several known classes of DNA photolyases in fungi, including the canonical CPD photolyase I family shown to directly repair cyclobutane dimers, as well as a cyr-DASH family shown to be involved in light-sensing phenotypes[52,53]. A maximum likelihood phylogeny shows that *P. destructans* VC83_00225 is a member of the cry-DASH family of DNA photolyases and is thus unlikely to contribute to direct photorepair of cyclobutane dimers (Fig. 5a). To be thorough, we tested photoreactivation in the laboratory and found no phenotype in *P. destructans*, while two of the nonpathogenic *Pseudogymnoascus* species that harbored at least one CPD photolyase I (*P. verrucosus* and *P. sp. 24MN13*) showed increased survival when exposed to UV-A (366 nm) light after DNA damage (Fig. 5b).

In fission yeast, UVE1 is a key component of the AER pathway; therefore, our data suggest that most UV-damaged DNA in *Pseudogymnoascus* is repaired through AER. A small percentage of repair can be attributed to photoreactivation in *P. verrucosus* and *P. sp. 24MN13*; however, this appears to not occur in *P. destructans* as it lacks a cyclobutane-dimer photorepair enzyme. The *P. destructans* genome harbors clear homologs for the major enzymes involved in NER pathway (Supplementary Data 22), and thus, more work is needed to determine why the general NER and BER pathways are unable to compensate for loss of AER in *P. destructans*. Nevertheless, the extreme sensitivity to UV-C light in *P. destructans* represents a genetic pathway that could be exploited for active management of WNS of bats. Currently, UV-A (366 nm) light is being used as a WNS noninvasive field diagnostic tool because the characteristic cupping skin lesions fluoresce under this wavelength[54]. While UV-A light tested under laboratory conditions had no effect on survival of *P. destructans* (Fig. 4), the use of UV-A as a diagnostic tool suggests that treating individual bats with a dosage of UV-C light is feasible. The relatively low dose (~10 mJ/cm$^2$) of UV-C light required to kill *P. destructans* conidia could be applied to bats in a few seconds of exposure from a portable light source. More work is needed to understand the physiological effect of UV-C light on bats; however, it is encouraging that treatment of mammal fungal skin/ nail infections with UV-C light has been used for dermatophyte fungi causing onychomycosis[55], as well as wound infections caused by *Candida albicans*[56].

## Discussion

WNS represents one of the most severe wildlife diseases ever recorded. Currently, there are no practical treatment options for containing the spread of WNS in North America and thus the fungus has moved rapidly across the continent. Survivor populations of affected bat species can still be found near the disease outbreak epicenter, so that there is hope that species extinction will be avoided, albeit with a prolonged recovery[57]. However, the long-term effects on ecosystems involving bats will not be understood for decades. Comparative genomics analyses presented here suggest that *P. destructans* is likely a true fungal pathogen of bats, evolving alongside Eurasian bat species for millions of years. The annotated genome of *P. destructans* in addition to several nonpathogenic closely related species provides a framework for understanding the pathobiology of WNS. The serendipitous discovery that *P. destructans* lacks a UVE1 homolog and is therefore extremely sensitive to pyrimidine dimer inducing DNA-damaging agents is a vulnerability that could be exploited for WNS management.

## Methods

**Growth conditions, DNA extraction, and sequencing**. Fungal strains were grown in liquid-stationary culture[58] for several days at 15 °C, mycelia were lyophilized, and subsequently genomic DNA (gDNA) was extracted as previously described[59]. High-molecular-weight gDNA was confirmed by gel electrophoresis. For sequencing on the Ion Torrent Personal Genome Machine (PGM), a random 400-bp size fractionated library was constructed using Ion Plus Fragment Library Kit (#4471257), templated using the Ion PGM Template OT2 400 Kit (#4479878), loaded on a 318v2 chip (#4484354, and sequenced using the Ion 400 bp Sequencing Kit (#4482002) (all kits were used according to the manufacturer's recommendations). Library construction for sequencing using the Illumina GAx II and Illumina MiSeq was done using the NEBNext Library Prep Kit, purified using QIAQuick Cartridge Kit, and size fractionated using E-gel 2% size select gel electrophoresis system (ThermoFisher). For obtaining RNA-seq reads to assist with genome annotation, fungi were grown in liquid-shaking cultures at 15 °C for 3 days, mycelia were lyophilized, total RNA was extracted using TriZol (Invitrogen), and poly-adenylated RNA was selected using the DynaBeads PolyA Cleanup Kit (Ambion). The poly-A RNA was then used to make libraries using the Ion RNA-seq Kit 2.0 and sequenced on the Ion Torrent PGM using Ion 200 bp Sequencing Kit (#4482006).

**Genome assembly**. Initial attempts to build a high-quality assembly for *P. destructans* using paired-end MiSeq (2 × 250 bp) data resulted in heavily fragmented assemblies with several different assembly software (Discovar DeNovo, Abyss, Spades, CLC, etc.). We generated a highly contiguous assembly using a combination of PacBio reads, MiSeq, Roche 454 mate-pair reads, and Sanger end sequences from a 100-kb BAC library[14]. For five of the nonpathogenic *Pseudogymnoascus* species (UAMH10579, 03VT05, 05NY08, WSF3629, and 23342-1-I1), acceptable assemblies were generated using the de novo assembler in CLC Genomics Workbench 7.5 (Qiagen) using paired-end MiSeq reads (2 × 300 bp). For *Pseudogymnoascus sp.* 24MN13, the paired-end Illumina (2 × 100 bp; GAIIx) data were found to have some bacterial contamination, thus, we generated a second sequencing library from a clean gDNA extraction for the Ion Torrent PGM, which was used for an initial assembly and scaffolding using the paired-end Illumina reads. Contamination was removed from the assemblies using DeconSeq[60] and was subsequently error corrected using Pilon[61]. Small repetitive contigs from each assembly were identified using Mummer[62] and were removed if they were 95% identical over 95% of their length with other contigs in the assembly (achieved using "funannotate clean" command).

**Genome annotation and functional characterization**. Detailed information on annotation and functional characterization is presented in Supplementary Methods.

**Biolog phenotyptic microarray**. *Pseudogymnoascus* species were tested for the ability to utilize 190 different carbon sources in the Biolog Phenotyptic Microarray platform (plates PM1 and PM2A). Fungal spores at a density of $1 \times 10^4$ spores were inoculated into 100 μl of carbon-free minimal medium (MM lacking glucose[63]) and then subsequently aliquoted into 96-well Biolog plates (PM1 and PM2A). Cultures were grown at 15 °C and the nonpathogenic *Pseudogymnoascus* species were incubated for 7 days, while *P. destructans* was incubated for 14 days. The plates were then photographed and growth was quantified using the Fiji/ImageJ plug-in ColonyArea 1.5[64]. Relative growth was then normalized to growth on glucose and plotted in heatmap.2 from the gplots package in R[65].

**DNA damage assays**. Conidia were harvested from fungal isolates from solid medium agar plates using sterile 0.01% Tween-80 water, purified by filtering over sterile miracloth (EMD Millipore #475855), and enumerated using a hemocytometer. For spot plate assays, conidia were serially diluted and 5 μl was pipetted onto the surface of glucose minimal agar medium[63]. Treatment with UV light was accomplished by exposing the spotted agar plate to 25 mJ/cm$^2$ of UV-C (254 nm) light using a UV cross-linker (UVP CL-1000). Sensitivity to chemical DNA mutagens was done by preparing GMM agar medium that contained 25 μM CPT, 0.01% MMS, or 0.5 μg/ml of 4-nitroquinolone (4-NQO). Control and treated plates were incubated at 15 °C for 1 week prior to imaging. Colony-forming unit (CFU) assays were done by spread plating ~50 conidia on the surface of a GMM agar plate and were subsequently exposed to UV light in a UV cross-linker, and surviving colonies were counted after incubation at 15 °C for 1 week. Quantification of photoreactivation for each species was done using a very similar CFU assay comparing varying UV-C treatments followed by 1-h exposure to UV-A (366 nm) light in a UV cross-linker. Conidia were allowed to germinate for 24 h in the dark prior to UV-C (254 nm) treatments and subsequent UV-A (366 nm) treatment. Surviving colonies were counted after incubation in the dark at an appropriate temperature (15 °C for *P. destructans* and 25 °C for nonpathogenic *Pseudogymnoascus* species). All CFU assays were done in biological triplicates ($n = 3$) for each experimental condition.

**Data availability**. Sequencing data and genome assemblies are available via NCBI under BioProject PRJNA276926 and SRA project SRP055906. All other relevant data supporting the findings of the study are available in this article and its Supplementary Information files, or from the corresponding author upon request.

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

## Acknowledgements

Funding was provided to J.M.P. and D.L.L. by the US Fish & Wildlife grant F14PG00132 and to K.P.D. and J.T.F. by US Fish & Wildlife grant F14AP00644. Additional funding and support for J.M.P. and D.L.L. were provided by the US Forest Service, Northern Research Station.

## Author contributions

J.M.P., K.P.D., J.T.F., and D.L.L. conceived the study; K.P.D. and J.T.F. performed whole-genome sequencing, K.P.D. conducted hybrid genome assembly of *P. destructans*, JMP conducted genome assembly, annotation, genome-level comparisons, and conducted laboratory experiments; and J.M.P. wrote the paper with input from K.P.D., J.T.F., and D. L.L.

## Additional information

**Competing interests:** The authors declare no competing financial interests.

