## [Peer Review File · Nature Communications]

Reviewers' comments:

Reviewer #1 (Remarks to the Author):

The study by Palmer et al. for the best part made use of published results to investigate the evolutionary history of bats infection by *Pseudogymnoascus destructans* (P.d.), which causes WNS. P.d. was compared to 6 closely related non-pathogenic *Pseudogymnoascus* species. The sequencing data indicated the expected genome size, GC content and a lower protein coding genome with 1934 unique proteins. It was found that P.d. has a low content of carbohydrate activating enzymes that is characteristic for fungal pathogens of animals. As previously described for other animal pathogens, the P.d.'s secretome was considerably reduced compared to those of non-pathogenic species. The authors reported additional characterization of P.d.'s biology - In general, this first set of data would be more suitable for specialized journals since it does not bring enough novelty to be considered for publication in a very high profile journal.

The authors suggest that P.d. co-evolved with bats in the absence of light. In the second set of experiments, the authors challenged the fungus to grow in the presence of DNA damage. The results indicated that P.d. was most sensitive to UV and MMS. In comparison with non-pathogenic strains, P.d. lacked UVE1 and it was proposed that UVE1 plays an important role in repair of UV induced DNA damage in P.d. Although interesting, these results do not explain why pyrimidine dimers are not efficiently removed by NER. Moreover in fungi like *S.cerevisiae*, that lacks UVDE, but also *S. pombe* pyrimidine dimers are efficiently removed by NER. Additionally, UVDE removes a number of DNA lesions that are repaired by BER. Is P.d. sensitive to these DNA lesions? For example, UVA is the major portion of sunlight and causes oxidative DNA damage that is repaired by BER. Thus, the results reported in the manuscript suggest that BER is efficient. To better understand the repair mechanisms, P.d. should be grown in the presence of DNA lesions that are removed by both, UVE and BER. UVA causes the formation of at least some pyrimidine dimers. Is P.d. resistant to UVA because the photolyase is activated? Photolyase is very active in removing pyrimidine dimers; it is present in microorganisms, fungi and plants. Is photolyase activity present and active in P.d.? In summary, the absence of UVE1 in P.d. is interesting but it remains an observation. Considerable more investigation is needed to explain the P.d. phenotypes in the presence of DNA damaging agents.

The authors suggested that treatment of bats with UV light is feasible. However, this is difficult to judge! It is stated that 5 to 10mJ/cm² (or 50 to 100 J/m²) UVC are a very low dose, but doses of ~15 J/m² are lethal for mammalian cells. In conclusion more experiments are needed to explain the sensitivity of P.d. to UVC and UVB but not to UVA. I am afraid that the results presented by Palmer et al., as they stand, are not of enough novelty to merit publication in an exceptionally high profile journal.

Reviewer #2 (Remarks to the Author):

The manuscript, "Extreme sensitivity to ultra-violet light in the fungal pathogen causing white-nose syndrome of bats" by Palmer et al conducted comparative study among 7 closely related fungal genomes including *P. destructans*, the causal agent of the Bat white-nose syndrome (WNS) that has caused bats decline in North American since 2006.

This is a well-written and nicely developed manuscript addressing an important problem. Here are some questions and suggestions.

1) The gem of this paper is the discovery of the sensitivity toward DNA damaging agents, as presented in figure 3. Obviously the other two strains *P. sp.* 23343 and *P. sp.* 3VT5 also have increased sensitivity as shown in the figure. Any genomic explanation for this?

2) Comparing to its close relatives, the bat pathogen has the largest genome, smallest number of protein-coding genes, largest proportion of repetitive sequences, but it also has the largest scaffold N50. This suggests a compartmentalized genome structure. The authors may check this reference (<https://www.nature.com/nature/journal/v464/n7287/full/nature08850.html>) to see any techniques they can use to check this possibility. The other suggestion is to create scaffold size histogram for all genomes to see whether there is a bimodal distribution for any their genomes. The authors also could check the affiliation of species-specific genes and the repetitive sequences.

3) Comparative study basically pointed toward a genome with shrinkage of functional gene families. The lack of enriched functional categories among *P. destructans*-specific genes could be an indication of our limited knowledge on fungal animal pathogens. I would suggest the authors to look into genes that perform function in specific host niche: such as higher body temperature, active host immunity, or other special nutrient sources.

4) To strengthen any argument presented in this paper is to confirm the observation in another *P. destructans* genome. I am surprised that the authors didn't cite the other *P. destructans* genome published at Genome Announcement, which is strongly suggested.

Reviewer #3 (Remarks to the Author):

This manuscript presents a thorough comparative genomics analysis of the fungal pathogen that causes white-nose syndrome in bats. The genome of *Pseudogymnoascus destructans* is shown to have several interesting features, when compared to six other species of closely related fungi that are not pathogenic. The authors conclude that these features demonstrate a long co-evolutionary history of the pathogen with its host. The most important conclusion may be that this fungus lacks a gene in a key DNA damage repair pathway and that it is unable to repair DNA from relatively low-level doses of short-wavelength UV radiation. The manuscript demonstrates the feasibility of this approach to kill the fungus and the authors speculate that it might be developed into a useful treatment or preventative for WNS.

This study is highly novel, will be of interest to everyone studying bats, fungal pathogens, and host-pathogen interactions. This paper will have a major impact on the thinking about WNS. More broadly, the implications for treating a disease that has devastated wildlife in North America will be of interest to all naturalists and environmentalists.

The bioinformatics and other analyses are well-performed and described clearly.

Although the manuscript is well-written, I believe that it could be improved by addressing the following minor points.

1. The statement in the abstract, "These genomic features, along with an estimation of last common ancestor at 23.5 MYA, indicate *P. destructans* has a long evolutionary history with and pathogenesis of bats" does not follow from the previous statement. The hypothesis that the loss of CAZyme genes contributes to pathogenicity is not well supported (see below) and this statement makes too direct a connection between the two.
2. The statement that "Most hibernating bats are nocturnal" is not relevant to the infection of *P. destructans* of bats because the infection only occurs during hibernation – the fungus is cleared within weeks of emergence from hibernation. The growth of the fungus in caves (and mines) and the infection of bats will occur in the dark, whether the bats are nocturnal or not.
3. The observation of reduced CAZyme genes and the correlation to reduced carbon source utilization is very interesting. However, more detail is needed to support the idea that this change is involved in the evolution of pathogenicity. The sentence on lines 152-154 should be placed before

the preceding one and additional explanation(s) should be added to further support the statement that "These data suggest an evolutionary trend towards pathogenicity."

4. Line 212: "or" should be "of"

5. Table 1 has the rows and columns switched. The different species should be rows to facilitate comparisons down each column. (I am also curious how the new assembly compares to

6. The authors fail to compare the genomic evidence that they provide to the transcriptomic work that has previously been published. This is a significant oversight as they mention two of the transcriptomic datasets were used to improve gene prediction on Line 555. When mapped back to their genome assembly are the gene expression results consistent with their conclusions about CAZymes, Subtilisins, and DNA repair enzymes?

7. Line 565 – "PGM" is undefined. Never mind, it is defined earlier, but it would be more clear to state "Ion Torrent PGM" here.

8. The manuscript is missing references in the Main Text that are directly relevant for the current study. Most glaring is the absence of the Drees et al Genome Announcements citation (it is found only in the supplemental material). The absence of this reference leads to the question of whether or not this is the same genome assembly for *P. destructans* or have the authors further refined the assembly for this manuscript? Extended Data Table 1 indicates that this is the same assembly, but in that case it should be more clear from the main text. Another important omission from the citation list (both the main and the supplementary) is Field et al PLoS Pathogens. This paper addressed some of the same questions (particularly about the subtilisin genes) from a transcriptomic perspective and the results should be compared to the current study.

Reviewer #4 (Remarks to the Author):

This is a carefully performed comparative genomic analysis of the hot-topic genus *Pseudogymnoascus*, comparing the pathogen Pd against its close non-pathogenic relatives.

For my comments to be taken in context, the authors should read our recent paper by Farrer et al in Nat Comms on the comparative genomics of two other pathogenic fungi (chytrids) of vertebrates compared against their saprophytic relatives. What astounded me here is that the two papers – which appear equally well done and are somewhat similar in their inception – have very different outcomes. This is likely largely down to the breadth of infection strategies that the Kingdom fungi have evolved – the chytrids are basal and separated by '00 millions of years from the ascomycetes described here.

In the chytrids, secreted proteases are massively expanded (compared to the saprophytes) compared to the decrease seen here in Pd. Pd also shows a reduced secretome (counter to what is described for chytrids) and reduction on CAZyme diversity (also counter to what is described for chytrids by Farrer et al). These 'inverse results' are worth thinking about and (briefly) integrating into the study. The model that I have in my mind is that Pd is undergoing a 'microbial minimalism' genome reduction process as has been described by Nancy Moran, whereas the chytrids have had to innovate to infect amphibians, hence their reliance on amplified gene-families. Possibly because the chytrids evolved earlier? Showing that this evolutionary flexibility to gaining pathogenicity exists across the fungal kingdom will add a lot to the study. I also have in my mind that studies in another ascomycete pathogen, *Coccidioides*, may show some of the trends found here - worth checking out Sharpton et al from 2009 and allied publications as there may be parallels there that you have missed.

As Pd appears to be on a trajectory to a reduced bat-associated genome (including reduction of secondary metabolites), I'd imagine that this will impact Pd's ability to persist as a saprophyte in cave environments. Do the authors agree and is this another chink in the pathogens armour? Does a probiotic approach to tackling WNS hold merit?

In the chytrids (sorry to harp on...) we found interesting expansions in the repeats. The Pd study doesn't really report anything on the repeats/transposons.. – is this because there is nothing of interest? Worth commenting on either way.

The serendipitous finding of UVE1 loss is persuasive and exciting – and adds an important potentially translational aspect to the study. I look forward to further experiments chasing this discovery.

L105 'it has been suggested'...

We have addressed the reviewers' comments in a point-by-point fashion and our responses can be found below, in bold.

Reviewer #1 (Remarks to the Author):

The study by Palmer et al. for the best part made use of published results to investigate the evolutionary history of bats infection by *Pseudogymnoascus destructans* (P.d.), which causes WNS. P.d. was compared to 6 closely related non-pathogenic *Pseudogymnoascus* species. The sequencing data indicated the expected genome size, GC content and a lower protein coding genome with 1934 unique proteins. It was found that P.d. has a low content of carbohydrate activating enzymes that is characteristic for fungal pathogens of animals. As previously described for other animal pathogens, the P.d.'s secretome was considerably reduced compared to those of non-pathogenic species. The authors reported additional characterization of P.d.'s biology - In general, this first set of data would be more suitable for specialized journals since it does not bring enough novelty to be considered for publication in a very high profile journal.

The assembly of *P. destructans* 20631-21 was indeed recently published in a Genome Announcement (Drees, et al. 2016); however, Genome Announcements is not a peer-reviewed journal and no comparative analyses were included in that work. All sequencing data, assemblies and annotations for these 7 fungal genomes have not been previously published. The genomes have been deposited in GenBank prior to publication and were thus publically available to provide the WNS research community with genomic resources as soon as possible. Therefore, a significant portion of this manuscript is devoted to describing these new genomes. Given the dramatic effects that WNS has had on North American bat populations and the surprising discovery that *P. destructans* is extremely sensitive to UV light, we feel (as did the other 3 reviewers) that this work is broadly interesting to the scientific community.

The authors suggest that P.d. co-evolved with bats in the absence of light. In the second set of experiments, the authors challenged the fungus to grow in the presence of DNA damage. The results indicated that P.d. was most sensitive to UV and MMS. In comparison with non-pathogenic strains, P.d. lacked UVE1 and it was proposed that UVE1 plays an important role in repair of UV induced DNA damage in P.d. Although interesting, these results do not explain why pyrimidine dimers are not efficiently removed by NER. Moreover in fungi like *S.cerevisiae*, that lacks UVDE, but also *S. pombe* pyrimidine dimers are efficiently removed by NER. Additionally, UVDE removes a number of DNA lesions that are repaired by BER. Is P.d. sensitive to these DNA lesions? For example, UVA is the major portion of sunlight and causes oxidative DNA damage that is repaired by BER. Thus, the results reported in the manuscript suggest that BER is efficient. To better understand the repair mechanisms, P.d. should be grown in the presence of DNA lesions that are removed by both, UVE and BER. UVA causes the formation of at least some pyrimidine dimers. Is P.d. resistant to UVA because the photolyase is activated? Photolyase is very active in removing pyrimidine dimers; it is present in microorganisms, fungi and plants. Is photolyase activity present and active in P.d.? In summary, the absence of UVE1 in P.d. is interesting but it remains an observation. Considerable more investigation is needed to explain the P.d. phenotypes in the presence of DNA damaging agents.

We do not yet fully understand the mechanism(s) for UV-induced DNA damage repair in *Pseudogymnoascus* given that very little is known about the biology of this group of fungi. While DNA repair pathways are hypothesized to be conserved in fungi, there is ample evidence in the literature to show that different fungi rely on different pathways to fix DNA damage via UV light. For example, the ascomycete yeast *S. cerevisiae* does not have UVDE, nor does *Candida albicans*; however, most filamentous fungi do have UVDE (UVE1) homologs. DNA repair of UV lesions in *Botrytis cinerea* relies largely on DNA photolyase activity because light is required for DNA repair. On the other hand, DNA photolyase activity in *A. nidulans* is masked by other repair pathways.

We were able to do another experiment in relation to the DNA photolyase repair ability of *P. destructans*. *P. destructans* does have a homolog of a DNA photolyase and to test if photoreactivation repair was functioning in *P. destructans* we challenged the fungus with UV light and then incubated in either 100% darkness or 100% light (full spectrum light source). This experiment, presented in Extended Data Figure 3, shows there is no differential survival of *P. destructans* under the light incubation compared to dark, which suggests that the DNA photolyase does not contribute substantially to DNA repair of UV-induced lesions. Since the DNA photolyase requires light for repair, it is unlikely to play a role in repair for *P. destructans*, considering the fungus lives primarily in dark environments. While DNA photolyases have been shown to be heavily involved in DNA repair in *Botrytis cinerea*, in other fungi, such as *Aspergillus nidulans*, the DNA photolyase (CryA) was shown to have no measurable effect on repair of UV damage despite the enzyme being functional when expressed heterologously in *E. coli*. This result led the authors to conclude that the repair effects of CryA in *A. nidulans* were masked by other DNA repair pathways such as NER and UVDE. In *P. destructans*, these data indicate that the DNA photolyase is either non-functional or insufficient to repair the damaged DNA under a 25 mJ/cm² exposure. These data support the notion that Pd is lacking UVE1 repair pathway and that levels of UV light used in this study cause DNA lesions that are unable to be sufficiently repaired by the NER pathway. This result has been added to the Extended Dataset and text has been modified on Lines 237 - 243 discussing this result.

Unfortunately, at this time no *Pseudogymnoascus* species are genetically tractable to the point where we have been able to make gene deletions. However, we are actively working on this technology, although this effort has been hampered by the slow growth of *P. destructans*. Thus, we are developing approaches for genetically manipulating this fungus, but it is a slow process and not a line of work that can be easily incorporated into the current manuscript.

The authors suggested that treatment of bats with UV light is feasible. However, this is difficult to judge! It is stated that 5 to 10mJ/cm² (or 50 to 100 J/m²) UVC are a very low dose, but doses of ~15 J/m² are lethal for mammalian cells. In conclusion more experiments are needed to explain the sensitivity of P.d. to UVC and UVB but not to UVA. I am afraid that the results presented by Palmer et al., as they stand, are not of enough novelty to merit publication in an exceptionally high profile journal.

While it is true that mammalian cells in tissue culture are very sensitive to UV light, we do not feel that is a fair or valid comparison. Mammalian skin is capable of withstanding much higher dosages of UV light than in tissue culture. In this study we exposed fungal spores to UV light – spores of fungi are protected by a fortified cell wall made specifically

for dispersal and long-term survival. Fungal spores can lay dormant for years and for many species are highly resistant to extreme environmental conditions such as UV light. UV-A (366 nm) light has rarely been shown to be anti-fungal and would require long dosages (i.e. 20 minutes to see a measurable effect in germination; however even after 4 hours some conidia still germinated in two filamentous fungi - Osman et al. 1989. Mycological Research).

In this manuscript we reference in the text (lines 252-253) that Dai et al used a dose of 2.92 – 6.48 J/cm² of UV-C light to treat mice infected with *Candida albicans*. The authors for that work report that skin damage occurred in mice; however, lesions recovered after 24 hours and this level of UC-C exposure was sufficient to kill 99.2% (2.92 J/cm² 30 min post- inoculation) and 95.8% (6.48 J/cm² 24 hours post-inoculation) of *C. albicans* cells. We do not know the capacity for bats to withstand UV-C light, but we believe that this is a valid research question for follow-up studies, which we are actively pursuing with researchers who specialize in bat physiology. If bat skin has a similar tolerance to UV light as the skin of mice, then the relatively low exposure of UV-C light required to kill *P. destructans* conidia may be an effective treatment option.

We agree that it is difficult to judge if UV light can be used as an effective treatment method until it is actually tested. We have begun setting up these experiments, but will not have results for more than a year. One of the major challenges in treating wildlife diseases is finding treatment options that are cheap, effective and “portable” (can be used in field conditions). In the case of treating bats, we mention that treating with UV lights is feasible as currently researchers are using long wave UV light to monitor WNS skin lesions; thus, it represents a potentially cheap and portable method to treat high value bats. Large-scale treatment of WNS-infected bats is tremendously challenging, with no approved treatment methods available.

Reviewer #2 (Remarks to the Author):

The manuscript, "Extreme sensitivity to ultra-violet light in the fungal pathogen causing white-nose syndrome of bats" by Palmer et al conducted comparative study among 7 closely related fungal genomes including *P. destructans*, the causal agent of the Bat white-nose syndrome (WNS) that has caused bats decline in North American since 2006.

This is a well-written and nicely developed manuscript addressing an important problem. Here are some questions and suggestions.

1) The gem of this paper is the discovery of the sensitivity toward DNA damaging agents, as presented in figure 3. Obviously the other two strains *P. sp* 23343 and *P. sp* 3VT5 also have increased sensitivity as shown in the figure. Any genomic explanation for this?

Fungi employ at least two "levels" of resistance to UV light, the first being structural/physical "barrier" where the cell wall can contain metabolites that absorb UV light, thereby physically protecting the cell from the radiation. The second level of resistance relies on cellular DNA repair machinery. Melanin is the most widely known/studied metabolite that is found in fungal cell walls that protects cells from UV light. Five of the seven species studied here contain a DHN-melanin secondary metabolite gene cluster (including *P. destructans*); however *P. 23342-1-I1* and *P. sp* WSF3629 do not contain this melanin cluster. It is been documented in other fungi that additional small molecules are involved in protection from UV light, so there are many other metabolites/pigments that a fungus could potentially be using for this purpose.

Perhaps the increased sensitivity of *P. sp* 23342-1-I1 and *P. sp* WSF3629 could be partially explained by lack of melanin or a related metabolite in the cell wall. However, *P. sp* 3VT05 does contain a DHN-melanin cluster (we do not know if it is functional) and thus we do not have a satisfying genomic explanation for this differential susceptibility as they all seem to contain what look to be functional DNA repair pathways. Of course, this is only based on predicted genome annotation, so more detailed work is needed in the future.

2) Comparing to its close relatives, the bat pathogen has the largest genome, smallest number of protein-coding genes, largest proportion of repetitive sequences, but it also has the largest scaffold N50. This suggests a compartmentalized genome structure. The authors may check this reference (<https://www.nature.com/nature/journal/v464/n7287/full/nature08850.html>) to see any techniques they can use to check this possibility. The other suggestion is to create scaffold size histogram for all genomes to see whether there is a bimodal distribution for any their genomes. The also also could check the affiliation of species-specific genes and the repetitive sequences.

These are great suggestions. In order to see if the 2,104 *P. destructans* genes that did not have orthologs in the non-pathogenic *Pseudogymnoascus* species (*Pd_uniques*) were physically located near/amongst repetitive sequences we did the following analyses.

1. Using repeatmasker output, we extended the repeat region windows by 1 kb on both sides of the intervals.
2. We then calculated the jaccard intersection statistic for the *Pd_uniques*, to 20 random samples of 2,104 proteins from the entire genome (*Pd_random*).
 - a. *Pd_uniques* = 4.69% overlap with repeat windows

- b. Pd_random = 5.27% (stdev = 0.099%) overlap with repeat windows
3. We then also calculated the relative distance between the unique genes and the repeats using bedtools reldist. The dotted line is the fraction that would be expected if the relative distances were uniformly distributed. Non-uniform distribution would be fractions greater than 0.05.

4. Taken together, these data suggest there is no correlation between the unique *P. destructans* genes and the repetitive sequences; thus in *P. destructans* there do not seem to be genomic pockets of species-specific regions.

3) Comparative study basically pointed toward a genome with shrinkage of functional gene families. The lack of enriched functional categories among *P. destructans*-specific genes could be an indication of our limited knowledge on fungal animal pathogens. I would suggest the authors to look into genes that perform function in specific host niche: such as higher body temperature, active host immunity, or other special nutrient sources.

Great suggestion. A few weeks ago, a study was published on the transcriptome dynamics of Pd during infection (Reeder et al. 2017. Virulence). The authors (J. Palmer is a co-author) concluded that there were 94 genes that were upregulated during WNS infection in comparison to growth in laboratory culture medium. We have cross-referenced the list of 94 upregulated genes with unique genes found in *P. destructans* and there are only 3 genes shared between these lists. All 3 of these genes are short hypothetical proteins with no known functional annotation. Thus, it does not appear that the unique proteins of *P. destructans* identified in this study are highly upregulated during WNS infection. In the transcriptome paper, the authors talk about genes that are differentially expressed; thus, we will not repeat those results here. From a genomics perspective, there does not appear to be expansions of protein families involved in the ability to cause WNS in bats. The reviewer also bring up a good point: we know very little about functional annotation in fungal genomes and thus it is difficult to predict function because the only available annotation is inferred from sequence homology. One of the reasons this manuscript will benefit the WNS community is because it will provide a genomics framework from which questions about WNS can be answered – such as the transcriptomics paper that was recently published.

4) To strengthen any argument presented in this paper is to confirm the observation in another *P. destructans* genome. I am surprised that the authors didn't cite the other *P. destructans* genome published at Genome Announcement, which is strongly suggested.

The Genome Announcement is cited in this manuscript and the assembly/annotation is

the same; however, due to citation limits by the journal the citation ended up in the extended data section. We have adjusted citations to include the genome announcement in the main text so there is no confusion regarding multiple assemblies/annotations. At this point we do not have multiple *P. destructans* genomes – there was a single introduction into North America (Lorch, Palmer, et al. mSphere doi:10.1128/mSphere.00148-16). We have some WGS information for a handful of European isolates, but not enough information to assemble a quality genome (due to the large amount of repetitive sequences). This is an active area of research.

Reviewer #3 (Remarks to the Author):

This manuscript presents a thorough comparative genomics analysis of the fungal pathogen that causes white-nose syndrome in bats. The genome of *Pseudogymnoascus destructans* is shown to have several interesting features, when compared to six other species of closely related fungi that are not pathogenic. The authors conclude that these features demonstrate a long co-evolutionary history of the pathogen with its host. The most important conclusion may be that this fungus lacks a gene in a key DNA damage repair pathway and that it is unable to repair DNA from relatively low-level doses of short-wavelength UV radiation. The manuscript demonstrates the feasibility of this approach to kill the fungus and the authors speculate that it might be developed into a useful treatment or preventative for WNS.

This study is highly novel, will be of interest to everyone studying bats, fungal pathogens, and host-pathogen interactions. This paper will have a major impact on the thinking about WNS. More broadly, the implications for treating a disease that has devastated wildlife in North America will be of interest to all naturalists and environmentalists.

The bioinformatics and other analyses are well-performed and described clearly.

Although the manuscript is well-written, I believe that it could be improved by addressing the following minor points.

1. The statement in the abstract, “These genomic features, along with an estimation of last common ancestor at 23.5 MYA, indicate *P. destructans* has a long evolutionary history with and pathogenesis of bats” does not follow from the previous statement. The hypothesis that the loss of CAZyme genes contributes to pathogenicity is not well supported (see below) and this statement makes too direct a connection between the two.

This sentence was changed.

2. The statement that “Most hibernating bats are nocturnal” is not relevant to the infection of *P. destructans* of bats because the infection only occurs during hibernation – the fungus is cleared within weeks of emergence from hibernation. The growth of the fungus in caves (and mines) and the infection of bats will occur in the dark, whether the bats are nocturnal or not.

Sentence changed.

3. The observation of reduced CAZyme genes and the correlation to reduced carbon source utilization is very interesting. However, more detail is needed to support the idea that this change is involved in the evolution of pathogenicity. The sentence on lines 152-154 should be placed before the preceding one and additional explanation(s) should be added to further support the statement that “These data suggest an evolutionary trend towards pathogenicity.”

Modified these sentences.

4. Line 212: “or” should be “of”

Thanks for catching.

5. Table 1 has the rows and columns switched. The different species should be rows to facilitate comparisons down each column. (I am also curious how the new assembly compares to

Table 1 was transposed.

6. The authors fail to compare the genomic evidence that they provide to the transcriptomic work that has previously been published. This is a significant oversight as they mention two of the transcriptomic datasets were used to improve gene prediction on Line 555. When mapped back to their genome assembly are the gene expression results consistent with their conclusions about CAZymes, Subtilisins, and DNA repair enzymes?

A few weeks ago, a paper was published on the transcriptome of *P. destructans* during WNS compared to growth in laboratory culture medium (J. Palmer is a co-author). More work is needed to look at transcriptome dynamics of DNA repair enzymes, and this is on our list of follow-up studies.

7. Line 565 – “PGM” is undefined. Never mind, it is defined earlier, but it would be more clear to state “Ion Torrent PGM” here.

Added.

8. The manuscript is missing references in the Main Text that are directly relevant for the current study. Most glaring is the absence of the Drees et al Genome Announcements citation (it is found only in the supplemental material). The absence of this reference leads to the question of whether or not this is the same genome assembly for *P. destructans* or have the authors further refined the assembly for this manuscript? Extended Data Table 1 indicates that this is the same assembly, but in that case it should be more clear from the main text. Another important omission from the citation list (both the main and the supplementary) is Field et al PLoS Pathogens. This paper addressed some of the same questions (particularly about the subtilisin genes) from a transcriptomic perspective and the results should be compared to the current study.

We apologize that it was not clear that the assembly was the same as in Drees. et al. We have now cited this genome announcement in the text so there is no confusion. As well we cited the Field et al. paper in the subtilisin protease discussion. Part of the problem here was this draft was written for Nature, which has a 50 citation limit. Given all the tools/software/etc used in a genomics project, it became difficult to fit in the necessary citations.

Reviewer #4 (Remarks to the Author):

This is a carefully performed comparative genomic analysis of the hot-topic genus *Pseudogymnoascus*, comparing the pathogen Pd against its close non-pathogenic relatives.

For my comments to be taken in context, the authors should read our recent paper by Farrer et al in Nat Comms on the comparative genomics of two other pathogenic fungi (chytrids) of vertebrates compared against their saprophytic relatives. What astounded me here is that the two papers – which appear equally well done and are somewhat similar in their inception – have very different outcomes. This is likely largely down to the breadth of infection strategies that the Kingdom fungi have evolved – the chytrids are basal and separated by '00 millions of years from the ascomycetes described here.

Fungi continue to amaze/impress with the different evolutionary histories that lead to a similar ecological outcome. This observation would perhaps make for an interesting post-publication commentary!

In the chytrids, secreted proteases are massively expanded (compared to the saprophytes) compared to the decrease seen here in Pd. Pd also shows a reduced secretome (counter to what is described for chytrids) and reduction on CAZyme diversity (also counter to what is described for chytrids by Farrer et al). These 'inverse results' are worth thinking about and (briefly) integrating into the study. The model that I have in my mind is that Pd is undergoing a 'microbial minimalism' genome reduction process as has been described by Nancy Moran, whereas the chytrids have had to innovate to infect amphibians, hence their reliance on amplified gene-families. Possibly because the chytrids evolved earlier? Showing that this evolutionary flexibility to gaining pathogenicity exists across the fungal kingdom will add a lot to the study. I also have in my mind that studies in another ascomycete pathogen, *Coccidioides*, may show some of the trends found here - worth checking out Sharpton et al from 2009 and allied publications as there may be parallels there that you have missed.

We agree with this analysis as well. We have added some of this into the discussion on Lines 190-201.

As Pd appears to be on a trajectory to a reduced bat-associated genome (including reduction of secondary metabolites), I'd imagine that this will impact Pd's ability to persist as a saprophyte in cave environments. Do the authors agree and is this another chink in the pathogens armour? Does a probiotic approach to tackling WNS hold merit?

Very good point/question. That is precisely our logic as well. We have preliminary data that show that Pd is likely not a good competitor in hibernacula soil environments. There was recently a paper from Christine Salomon's group suggesting a similar observation (<http://journals.plos.org/plosone/article?id=10.1371/journal.pone.0178968>). We are actively following up on this as well, and have growth experiments in process with sterile/non-sterile soil.

In the chytrids (sorry to harp on...) we found interesting expansions in the repeats. The Pd study doesn't really report anything on the repeats/transposons.. – is this because there is nothing of interest? Worth commenting on either way.

We have looked at repeats in the genomes and there is a remarkable expansion of repeats in *P. destructans*: nearly 40% of the genome is repetitive. However, we are still looking into characterizing these repeats and trying to generate more data along these lines. We are sequencing additional Eurasian isolates of *P. destructans* with the goal of a follow up paper characterizing repeats and looking at distributions of repetitive elements in the several isolates of Pd from Eurasia. The preliminary data suggest there are some genome rearrangements in the regions of high repeat density. We did not feel

there was enough space to accurately describe the repeat story here in addition to the UV story. Since the UV story has management implications, we feel it is critical to get these data out first and then follow-up with a more comprehensive genome rearrangement/repeat manuscript. We have added a brief mention of the expanded repeats in connection to the genome reductionism in some protein families for some commentary on comparison to other fungal pathogens as suggested.

The serendipitous finding of UVE1 loss is persuasive and exciting – and adds an important potentially translational aspect to the study. I look forward to further experiments chasing this discovery.

Thank you, we hope to do some laboratory UV trials during the next hibernation season.

L105 'it has been suggested'...

Fixed this error, thanks.

Reviewer #1 (Remarks to the Author):

In general, I am satisfied by the modifications that were made to the manuscript, although I think that more information on the inefficiency of both, photolyase and NER should and could have been provided.

Before the final acceptance of the manuscript, I advise the authors to make the following changes:

- 1) Delete 'Extended Data Figure 3'; the results are too preliminary. Different and measured doses of near-UV/blue light (300-500 nm) should have been used. Both, *P. destructans* and non-pathogenic *Pseudogymnoascus* should have been treated in parallel with increasing doses of UVC, as done in Fig. 3A.
- 2) Change all 'low doses' in the text to 'relative low doses'
- 3) P.9: 'UV disrupts DNA...'; change to: 'UV light damages DNA by inducing the formation of...'
- 4) P.9: 'To combat DNA damage...'; change to: 'To remove DNA lesions...'
- 5) P.10: '....suggesting that UVE1 plays a large role in the repair of UV-damaged DNA in *Pseudogymnoascus*.'; here indicate the name of the gene coding for photolyase and names of genes coding for all NER enzymes.
- 6) P.10: Delete the sentence: 'Laboratory testing(Extended Data Figure 3)': photolyase is specific for CPDs or 6-4PDs. If *P. destructans* has the CPD-photolyase, then 6-4PDs are not removed, which could inhibit the formation of colonies.

Reviewer #2 (Remarks to the Author):

Although I would like to see a scatterplot between scaffold size and the repeat content or a histogram of scaffold length distribution, with the understanding that this group didn't generate the initial pathogenic strain genome assembly, I can accept the paper as it is now.

Reviewer #4 (Remarks to the Author):

[No further comments for author.]

***** Responses in BOLD *****

Reviewer #1 comments:

Before the final acceptance of the manuscript, I advise the authors to make the following changes:

- 1) Delete 'Extended Data Figure 3'; the results are too preliminary. Different and measured doses of near-UV/blue light (300-500 nm) should have been used. Both, *P. destructans* and non-pathogenic *Pseudogymnoascus* should have been treated in parallel with increasing doses of UVC, as done in Fig. 3A.
- 2) Change all 'low doses' in the text to 'relative low doses'
- 3) P.9: 'UV disrupts DNA...'; change to: 'UV light damages DNA by inducing the formation of...'
- 4) P.9: 'To combat DNA damage...'; change to: 'To remove DNA lesions....'
- 5) P.10: '...suggesting that UVE1 plays a large role in the repair of UV-damaged DNA in *Pseudogymnoascus*.'; here indicate the name of the gene coding for photolyase and names of genes coding for all NER enzymes.
- 6) P.10: Delete the sentence: 'Laboratory testing(Extended Data Figure 3)': photolyase is specific for CPDs or 6-4PDs. If *P. destructans* has the CPD-photolyase, then 6-4PDs are not removed, which could inhibit the formation of colonies.

The minor points above have been changed in the text. We also conducted a more thorough investigation of the putative DNA photolyases in *P. destructans* and the *Pseudogymnoascus* species studied in this manuscript. We found that the *P. destructans* DNA photolyase is part of the *cyr-DASH* family, which has been shown to be involved in light sensing in other filamentous fungi, although not involved in direct photoreactivation DNA repair. Newly completed photoreactivation assays confirmed this phenotype, where exposure to UV-A light for 1 hour did not increase survival of *P. destructans*; however, two of the other *Pseudogymnoascus* species that do harbor a CPD Photolyase I did show an increase in survival under the same conditions (Extended Data Figure 5). Since WNS occurs in hibernacula and in the absence of light, our general conclusion remains the same, where it appears that loss of UVE1 is likely the reason that *P. destructans* is extremely sensitive to UV-C light.

Reviewer #2 (Remarks to the Author):

Although I would like to see a scatterplot between scaffold size and the repeat content or a histogram of scaffold length distribution, with the understanding that this group didn't generate the initial pathogenic strain genome assembly, I can accept the paper as it is now.

We apologize that the previous data were not sufficient to address this issue. We have added a reference to this in the text as well as Extended Data Figure 3 – which shows that there are not pockets of repetitive elements in the *P. destructans* genome assembly. This is in contrast to a phenomenon observed in *Fusarium oxysporum* where this pathogen harbors min-chromosomes that contain a high percentage of repeats as well as lineage specific genes that have subsequently been shown to be involved in pathogenicity. Moreover, the lineage-specific genes are not co-localized with repetitive elements in the genome. It seems rather that the repeat content of the genome is high and fairly uniform across the chromosomes.

Reviewer #4 (Remarks to the Author):

[No further comments for author.]